# Conjugative Plasmid pPPUT-Tik1-1 from a Permafrost *Pseudomonas putida* Strain and Its Present-Day Counterparts Inhabiting Environments and Clinics

**DOI:** 10.3390/ijms241713518

**Published:** 2023-08-31

**Authors:** Olga Maslova, Alexey Beletsky, Sofia Mindlin, Nika Petrova, Andrey Mardanov, Mayya Petrova

**Affiliations:** 1National Research Centre “Kurchatov Institute”, 123182 Moscow, Russia; elvi.23@mail.ru (O.M.); nikafpetrova@gmail.com (N.P.); 2Institute of Bioengineering, Research Center of Biotechnology of the Russian Academy of Sciences, 117312 Moscow, Russia; mortu@yandex.ru (A.B.); mardanov@biengi.ac.ru (A.M.)

**Keywords:** novel group of plasmids, plasmid backbone, accessory region, antibiotic resistance, heavy metal resistance, conjugation

## Abstract

A novel group of conjugative plasmids of *Pseudomonas* is characterized. The prototype plasmid pPPUT-Tik1-1 (153,663 bp), isolated from a permafrost strain of *P. putida* Tik1, carries a defective mercury transposon, Tn*501*, and a streptomycin resistance transposon, Tn*5393.* Ten plasmids and 34 contigs with backbone regions closely related to pPPUT-Tik1-1 have been found in GenBank. Two of these plasmids from clinical strains of *P. putida* and *P. fulva* are almost identical to the ancient plasmid. A characteristic feature of this group of plasmids is the presence of two genes encoding the initiators of replication (*repA1* and *repA2*). None of these genes have high similarity with plasmid replication genes belonging to known incompatibility groups. It has been demonstrated that while pPPUT-Tik1-1-like plasmids have homologous backbone regions, they significantly differ by the molecular structure and the predicted functions of their accessory regions. Some of the pPPUT-Tik1-1-related plasmids carry determinants of antibiotic resistance and/or heavy metal salts. Some plasmids are characterized by the ability to degrade xenobiotics. Plasmids related to pPPUT-Tik1-1 are characterized by a narrow host range and are found in various species of the *Pseudomonas* genus. Interestingly, we also found shorter plasmid variants containing the same replication module, but lacking conjugation genes and containing other structural changes that strongly distinguish them from plasmids related to pPPUT-Tik1-1, indicating that the structure of the replication module cannot be used as the sole criterion for classifying plasmids. Overall, the results suggest that the plasmids of the novel group can be spread using conjugation in environmental and clinical strains of *Pseudomonas* and may play diverse adaptive functions due to the presence of various accessory regions.

## 1. Introduction

Strains of the *Pseudomonas* genus are found in most soil and aquatic environments and some are implicated in diseases of humans, animals and plants [1]. Many *Pseudomonas* isolates harbor plasmids, which contribute to the adaptability of *Pseudomonas* species in a variety of natural habitats [2,3,4].

The number of sequenced plasmids has greatly increased in recent years, necessitating the development of new approaches for their classification. Initially, classification of plasmids was based on their ability/inability to coexist in a bacterial cell [5,6,7]. According to this criterion, 14 groups of incompatibility were identified and described in *Pseudomonas* [6,8], and the role of replication initiation genes in this ability was shown [8,9]. Further studies of the molecular structure of plasmids, in particular their replication genes, and analyses of newly sequenced plasmids, required the development of novel approaches to improve their classification [8]. Solving this problem is difficult for a number of reasons, since some plasmids could not be classified on the basis of their incompatibility [10]. In particular, for plasmids of plant pathogens from the group of *P. syringae*, the ability to propagate together in the same cell is a characteristic feature [2]. The nucleotide sequences of the replication systems of plasmids belonging to IncP-5, IncP-8, IncP-10, IncP-11, IncP-12, IncP-13 and IncP-14 were not determined [8,11]. Furthermore, many plasmids remain outside of distinct incompatibility groups. For example, novel incompatibility plasmid groups, Inc_pRBL16_, have been recently described [12].

To improve the plasmid classification, two main methods have been developed later, i.e., typing for the replication initiation genes and typing for the relaxation (MOB genes, encoding the conjugative complex) genes [8,13,14,15,16,17]. While replicon typing aims at plasmid replication regions with non-degenerate primers, MOB typing classifies plasmids into relaxase subfamilies using degenerate primers. As a result, MOB typing provides a deeper phylogenetic depth than replicon typing [14]. The use of both methods resulted in a significant expansion of the number of detected new groups of plasmids [18,19].

Currently, there is no generally accepted classification system for *Pseudomona*s plasmids. In most cases, the criteria used for classification include the molecular structure of the backbone region (replication, partition, mobilization and conjugation modules) and the properties determined with the accessory region (resistance to heavy metals and antibiotics, ability to degrade xenobiotics and pathogenicity in relation to plants, animals and humans), as well as plasmid host range. At the same time, it is obvious that development of a satisfactory classification of plasmids is still far away [16,17].

In the present work, we describe a novel group of *Pseudomonas* plasmids. Its prototype plasmid was found in a *P. putida* Tik1 strain previously isolated from a permafrost sample aged 220–390 thousand years [20]. *P. putida* is a ubiquitous rhizosphere saprophytic bacterium and soil colonizer that belongs to the wide group of fluorescent Pseudomonas species [21]. An analysis of the permafrost strain, which has never come into contact with modern environmental or clinical strains, allowed us to determine the structure of the prototype pPPUT-Tik1-1 plasmid from this group that existed long before the beginning of active anthropogenic impact on the biosphere. The plasmid pPPUT-Tik1-1 is a large iteron-containing conjugative plasmid containing genes of mercury and streptomycin resistance. Other plasmids of the novel group, while containing the common backbone region, are characterized by various accessory regions and are spread among both environmental and clinical strains of the *Pseudomonas* genus. 

## 2. Results

### 2.1. The Prototype Plasmid pPPUT-Tik1-1 

Bacterial strain Tik1 was isolated from a permafrost sample collected in the area of the Laptev Sea Coast [20]. According to the phenotypic features, it was preliminary assigned to the species *P. putida.* The strain was resistant to inorganic mercury compounds and to streptomycin [20]. It was shown with conjugation mating that both mercury resistance and streptomycin resistance determinants are located on a large conjugative plasmid [20]. For a more detailed study of the ancient plasmid and its host strain, we performed whole-genome sequencing of the Tik1 strain. The analysis of the chromosomal sequence of Tik1 showed that this strain really belongs to the *P. putida* species (99.27% and 99.4% similarity with the *P. putida* strains JCM 18798 (BBDC00000000) and KF715 (NC_021505.1), respectively). 

The plasmid size, determined with its sequencing, is 153,663 bp; a schematic map of pPPUT-Tik1-1 is presented in Figure 1. The backbone region of pPPUT-Tik1-1 contains genes *repA1* and *repA2* encoding replication initiator proteins, operon *parAB* encoding the partition complex for segregation of plasmid copies and *mob*-*tra1*, *tra2* and *trb* modules encoding functions of plasmid conjugal transfer. The accessory region including transposons ΔTn*501* and Tn*5393,* containing resistance determinants to mercury compounds and streptomycin, respectively, is located between the *repA1* and *repA2* genes. Another group of accessory genes can be found between *trb and tra2* (Figure 1). It should be noted that pPPUT-Tik1-1 contains more insertions of various mobile elements, as well as their remnants, compared to the previously described related plasmid 1269-2 [22].

Interestingly, we revealed in GenBank two plasmids isolated in China, which were highly similar to pPPUT-Tik1-1. The first of them, p420352-strA (MT074087), found in a *P. putida* strain 15140 352 isolated in 2015, was an almost identical copy of pPPUT-Tik1-1. It differed from pPPUT-Tik1-1 by only a few features: an insertion of 15 bp in the gene *merA* and three single-nucleotide changes in the genes *repA1* and *mobA* and in the gene-encoding flavodoxin reductase. The second almost identical plasmid (unnamed, CP064947.1) was found in a clinical strain of *P. fulva* ZDHY14 (150,273 bp) isolated in 2019 from the pleural fluid in a respiratory ICU. In comparison with pPPUT-Tik1-1, it contained a 3401 bp deletion eliminating three genes encoding xanthine–uracil permease, nucleoside-binding outer membrane protein and 2-oxobutyrate oxidase. 

The discovery of slightly different variants of the same plasmid in ancient and modern *Pseudomonas* strains belonging to different species indicates the success of this plasmid and suggests its adaptability to various habitats.

### 2.2. Genome of pPPUT-Tik1-1 and Related Plasmids

In addition to the two previously mentioned plasmids, fourteen plasmids containing the same replication and maintenance genes (*repA1*, *repA2* and *parA-parB*) as pPPUT-Tik1-1 and therefore related to it were found by us in GenBank (Figure 2 and Appendix A and Table 1 and Appendix A). Ten of them contained conjugative genes and their backbone regions that were overall similar to pPPUT-Tik1-1 (Figure 2 and Appendix A). However, most of them differed significantly from pPPUT-Tik1-1 and from each other in the size and the structure of their accessory regions (Figure 2). 

The other four plasmids completely lacked conjugation genes. However, the remaining part of their backbone regions containing replication and maintenance genes was similar to the corresponding regions of conjugative plasmids.

We also found another plasmid (pGYL5) encoding the RepA1 protein that was only 56% similar to the RepA1 proteins encoded by the other 14 plasmids (Appendix A). Therefore, according to the existing classification it does not belong to the group of plasmids related to pPPUT-Tik1-1. The cluster analysis of all genes encoded by the 15 plasmids using the cd-hit program with a minimum global sequence identity of 50% with amino acids showed that conjugative and non-conjugative plasmids form two separate clusters, the members of each of which are similar in their structure as a whole (Appendix A). Based on this analysis, the pGYL5 plasmid is a member of the subgroup of non-conjugative plasmids, although it contains a different allele of the *repA1* gene.

The conjugative plasmids from the group of pPPUT-Tik1-1 have many more genes in common than non-conjugative ones. All these genes are found in at least nine plasmids from eleven (Figure 2). At the same time, some of them are most likely not directly related to the maintenance of plasmids, for example, the arabinose efflux permease gene and an orphan zeta gene belonging to the Zeta-Epsilon family [23]. The common backbone region of plasmids related to pPPUT-Tik1-1 is about 70 kb in length (72,289 bp in pPPUT-Tik1-1). The size of the smallest plasmid of this group, pS7-1M, which almost entirely consists of genes common to all plasmids of this group, is 68,717 bp. The pS7-1M plasmid contains only one gene that is not included in the backbone region. The other plasmids have many more additional genes. It should be noted that in most plasmids of this group, the common backbone regions include insertions of individual unique genes or small gene clusters (Figure 2). For example, the common backbone region of pPPUT-Tik1-1 includes eight such genes. The acquisition of all these genes is not associated with insertions of any mobile elements (Figure 1).

It is important to emphasize that a characteristic feature of pPPUT-Tik1-1-related plasmids is the presence of two genes encoding two different replication initiation proteins RepA1 and RepA2. Both genes are widely distributed in plasmids of various *Pseudomonas* species. Many plasmids from other groups (for example, pB13-200A from *P. syringae* pv. tomato B13-200 [CP019872.1]; pPHE20 from *P. fluorescens* PC20 [KY503036]; and pOXA-198 from *P. aeruginosa* PA41437 [MG958650.1]) contain only the larger *repA1* gene, while we did not find plasmids with a single *repA2* gene. Therefore, it can be suggested that the product of the *repA1* gene plays the main role in the process of replication initiation. 

This assumption was supported by the analysis of the structure of plasmid regions adjacent to the *repA1* genes. Earlier, Xu et al. [22] revealed in the plasmid p1269-2 of *P. putida* the eight 15-bp imperfect direct repeats of TCGTAtaTCaCCGAt, designated R1 to R8, upstream of the *repA1* gene, which were assigned as putative iterons. 

In the present study, we found that p1269-2 is a member of the group of plasmids related to pPPUT-Tik1-1. Therefore, we studied the structure of similar regions in pPPUT-Tik1-1 and in all related plasmids. In all plasmids containing conjugation transfer genes, the structure of this region was the same and coincided with that described by Xu et al. [22] (Figure 3a). The only exception in this group was the smallest plasmid pS7-1M, which lacked the third iteron, although the distance between the second and fourth iterons was the same as in other plasmids. In four smaller non-conjugative plasmids, the third iteron was also absent, but the distance between the second and fourth iterons was much larger than in the large plasmids, 1788 bp instead of 119 bp (Figure 3b). This resulted from the insertion of an IS*3* family element between them (Figure 3b). Thus, in non-conjugative plasmids, only five out of eight iterons can be involved in the replication initiation. The distances between these iterons and the structure of the downstream region were the same for all plasmids.

In contrast, no iteron-like sequences could be found upstream to the *repA2* gene. This further suggests that the gene *repA1* of pPPUT-Tik1-1-like plasmids is the main gene involved in the plasmid replication. 

### 2.3. Identification of the Replicon and Relaxase Types of pPPUT-Tik1-1 and Related Plasmids

According to modern approaches to plasmid classification, the replicon- and relaxase-based typing provide the most reliable results [8,13,14,16,24]. We used both approaches to characterize the plasmid pPPUT-Tik1-1 and related plasmids. When typing replicons, plasmid sequences of different incompatibility groups were used as references, since they directly reflect the structural features of replication proteins. Both replication genes of pPPUT-Tik1-1 were analyzed. The greatest similarity with the *repA1* gene of pPPUT-Tik1-1 was observed in plasmids of the recently described IncP-11 group [11] (Appendix A). However, the similarity of the *repA1* gene with the replication genes from plasmids of this group is only 72% and coverage is 88–96%. The similarity of the gene *repA2* with known replication genes is even more remote (Appendix A). Therefore, plasmids related to pPPUT-Tik1-1 could not be attributed to any known group of replicons and should be allocated to a new group.

It is noteworthy that based on the similarity of replication proteins, all plasmids from the pPPUT-Tik1-1 group, including eleven conjugative plasmids, and four shorter non-conjugative plasmids should be attributed to the same group (Table 1). However, the structure of the *tra1*, *tra2*, *trb* and *mob* loci was highly similar in eleven conjugative plasmids, suggesting that they can also be used for classification of pPPUT-Tik1-1 and related plasmids. Earlier, Xu et al. [20], who described the first plasmid from this group, identified its relaxase as a member of the MOB_P12_ clade. We performed a phylogenetic analysis of relaxases for plasmids related to pPPUT-Tik1-1, also taking for the analysis proteins encoded by *mob* genes from the MOB clades P11, P12, P13 and P14 [13]. This analysis revealed that the Mob protein encoded by pPPUT-Tik1-1 belongs to the MOB_P13_ clade (Appendix A). The *tra1*, *tra2* and *trb* loci belong to the T4SS secretion system. Therefore, for reliable qualification of plasmids, it is necessary to take into account the structure of all components of the backbone region, in particular, the RepA1, RepA2 and MobA proteins. 

### 2.4. Coevolution of the Main Genes of the Backbone Region

Our analysis revealed that plasmids having replicons related to pPPUT-Tik1-1 differ greatly in their structure. The loss of conjugation genes could occur only once or several independent times during the evolution of plasmids from the pPPUT-Tik1-1 group. To understand this issue, we performed a phylogenetic analysis of three key proteins from the backbone region of plasmids, both replication proteins RepA1 and RepA2 and the MobA relaxase. All three phylogenetic trees have a similar topology, which indicates co-evolution of all three proteins (Figure 4). Therefore, the loss of the *tra* region occurred only once during the evolution of this group of plasmids. Interestingly, the replication genes of one of the conjugative plasmids, pNY7610-IMP, are closer in structure to the genes of small non-conjugative plasmids. It is noteworthy that this plasmid was found in a strain of *P. aeruginosa*, the species that is most phylogenetically remoted from the other host plasmid strains from the pPPUT-Tik1-1 group.

### 2.5. Determination of the Conjugative Host Range of pPPUT-Tik1-1 and Its Stability in Different Hosts

Previously, we showed that the plasmid pPPUT-Tik1-1 is characterized by a narrow host range and could be transferred by conjugation to *Pseudomonas* strains but not to *E. coli* and *Acinetobacter* sp. [20]. In the present research, we analyzed transfer of plasmids related to pPPUT-Tik1-1 in various *Pseudomonas* species. By the beginning of this study, all plasmids from this group deposited in GenBank were found among strains belonging to the closest groups of *P. putida* and *P. fluorescens*. Only recently, one plasmid from this group was found in *P. aeruginosa* (Table 1). Therefore, we initially assumed that the plasmids from this group could have a narrow host range within closely related species in the *Pseudomonas* genus. To test this assumption, we performed conjugation crosses of the strain Tik1, the original host of pPPUT-Tik1-1, and typical strains of different *Pseudomonas* species belonging to different groups (Table 2). It was found that the strains of the remote group *P. aeruginosa* were unable to maintain the replication of pPPUT-Tik1-1 (the conjugation frequency of <10^−9^) (Table 2). At the same time, the frequency of transfer of pPPUT-Tik1-1 into strains belonging to more phylogenetically close groups, including *P. resinovorans, P. oleovorans, P. putida* and *P. fluorescence,* was 10^−3^–10^−6^ (Table 2). The plasmid pPPUT-Tik1-1 did not transfer into a strain of *P. seleniipraecipitans*. Probably, this may be due to some specific properties of the recipient strain [25]. Thus, the results of the conjugation mating experiments suggested that the plasmid can be transferred only into phylogenetically close species, confirming our assumption about a narrow host range of pPPUT-Tik1-1. 

However, a plasmid, pNY7610-IMP, related to pPPUT-Tik1-1, was isolated from *P. aeruginosa* (CP096914) (see above), and we also found four unassembled genomes of different environmental and clinical *P. aeruginosa* strains containing related plasmids (Appendix A). This suggests that plasmids from this group can also be maintained in other species of the *Pseudomonas* genus.

An important characteristic that must be considered during an analysis of the host range of plasmids is their stability in the absence of selective pressure [28]. Therefore, we tested the stability of the plasmid in recipients for more than 10 passages. During this experiment, 100% of cells retained the plasmid after passaging, independent of the phylogenetic position of the recipient, indicating high stability of the plasmid in the bacterial culture.

### 2.6. The Accessory Region of pPPUT-Tik1-1 and Related Plasmids

The accessory regions of plasmids from the pPPUT-Tik1-1 group are characterized by considerable diversity in their size (from 3.5 kb to 80 kb) and the adaptive functions encoded by them (Table 1). And among the adaptive features encoded by accessory genes in plasmids of this group are resistance to heavy metals and antibiotics and toluene biodegradation (Table 1 and Appendix A).

#### 2.6.1. Resistance to Heavy Metals

Many of the plasmids from the pPPUT-Tik1-1 group (including the 34 plasmids revealed in unassembled genomes) contained mercury resistance determinants. At the same time, determinants of resistance to copper and zinc/cobalt/cadmium (*czc* operon) were found in only one contig; two contigs contained determinants of chromium resistance (*chrAB*) (Appendix A).

We paid special attention to the study of mercury resistance operons and transposons present in the prototype plasmid pPPUT-Tik1-1 and other plasmids of this group. The plasmid pPPUT-Tik1-1 and its variants (p420352-strA and an unnamed plasmid from *P. fulva*) contained the defective transposon Tn*501*. It contained a full *mer* operon, almost identical to the *mer* operon of Tn*501,* and remnants of the Tn*501* transposition module. In particular, only the sequences of the left repeat, the spacer between it, the start of the *merR* gene and 43 bp from the *res* site were preserved. This structure likely arose as a result of site-specific recombination in the *res* site.

Unlike pPPUT-Tik1-1, the plasmid pCN1276 contained two complete mercury transposons, Tn*512-*like (Tn*50*5*3* family) and Tn*5046*-like (Tn*3* family); ΔTn*5041*D was revealed in the plasmid p12969-2 (Table 3). The same transposons were revealed in several contigs (Appendix A). Given the limited number of plasmids in this study, this suggests the high frequency of mercury resistance transposons and their high diversity in plasmids in various *Pseudomonas* strains, including clinical strains. 

Unexpectedly, we found another transposon (Tn*5563a*) in pPPUT-Tik1-1, containing an incomplete *mer* operon (*merRTP*), described by Yeo et al. [29]. It turned out that this transposon also occurs with high frequency in plasmids from the pPPUT-Tik1-1 group (in 16 from 45). In a number of cases, a complete *mer* operon was present in the same plasmid simultaneously with Tn*5563a* (Table 2 and Appendix A). 

#### 2.6.2. Biodegradation

Plasmid pGRT1 and contig node 6 contain genes (ttgGHI) encoding tolerance to toluene, and the pGRT1 was shown to provide the ability to use organic solvents for cell growth [30]. In both cases, the hosts of the plasmids were environmental strains of *Pseudomonas putida* (Table 2 and Appendix A).

#### 2.6.3. Antibiotic Resistance

Despite the fact that the vast majority of *Pseudomonas* plasmids (contigs) from the pPPUT-Tik1-1 group were isolated from clinical strains, only a few of them contained genes for resistance to antibiotics (4 out of 45). Thus, plasmid pPPUT-Tik1-1 and p12969-2 contained genes for resistance to streptomycin; pNY7610-IMP contained genes for resistance to streptomycin, gentamicin, kanamycin, carbapenems and cefalosporins; and contig from strain *Pseudomonas* sp. NM1542 contained two copies of the *bla*_VIM-2_ gene encoding resistance to beta-lactams. The absence of antibiotic resistance genes in the plasmids of clinical strains likely suggests that such genes can be located in other plasmids or in the chromosome; for instance, the strain *P. juntendii* 18091276 isolated from urine contained in the chromosome gene *bla*_IMP_ encoding resistance to beta-lactams while its plasmid pCN1276 lacked antibiotic resistance genes.

## 3. Discussion

In this study, we analyzed a novel group of pPPUT-Tik1-1-related plasmids, using previously developed criteria for plasmid classification based on the genetic organization, homology and size of the backbone regions [19]. As a result, a novel group of conjugative *Pseudomonas* plasmids was discovered. Along with pPPUT-Tik1-1 and eight other completely sequenced plasmids not previously described, it included two known plasmids: (i) p12969-2 from a clinical strain of *P. putida* [22] and (ii) pGRT1 from an environmental strain of *P. putida* [30]. The comparative analysis of the structure of all plasmids confirmed that they all belong to the same group, characterized by the presence of two replicative proteins RepA1 and RepA2. Despite significant differences in the size and structure of the accessory region, their genes encoding the main replicative and transfer functions (replication, partitioning, mobilization and conjugation) are characterized by high levels of homology (Table 1). Furthermore, the analysis of the phylogenetic trees was built for both replication initiation proteins and the relaxase MobA indicated co-evolution of the genes from the backbone region encoding different functions. This may indicate coordinated functioning of those genes during plasmid replication.

Based on the example of the *Pseudomonas* plasmid, we demonstrated the usefulness of a complex approach for plasmid classification, which takes into account the types of both replication and mobilization genes. The use of the classical approach to classify *Pseudomonas* plasmids by the type of their replicon proteins fails to separate two subgroups of the plasmids, containing or lacking the conjugation genes. Earlier, we found the same problem during the study of *Acinetobacter* plasmids [17]. The use of a complex approach in the classification of plasmids related to pPPUT-Tik1-1 made it possible to reliably determine the boundaries of the pPPUT-Tik1-1 group and separate the two subgroups of these plasmids. However, when considering various groups of plasmids, the situation is complicated by the fact that, on the one hand, many plasmids either do not contain a replication initiation gene, or contain two such genes, and on the other hand, not all plasmids contain mobilization genes. According to Orlek et al. [14], only 85% and 65% of the collected plasmids contain the replicon and MOB genes, respectively. Obviously, in the absence of these genes, it is necessary to develop specific approaches for different cases.

The prototype of the novel plasmid group, the plasmid pPPUT-Tik1-1, was isolated from a permafrost sample aged 220–390 thousand years. This conjugative plasmid possesses a complete set of determinants of the type IVSS conjugation system [31] and operon *parAB* encoding the partition proteins for segregation of plasmid copies during the cell-division process [32] and contains two genes encoding replication initiation proteins, one of which is adjacent to eight 15-bp iterons. Among plasmids that belonged to this group, there were plasmids found in environmental *Pseudomonas* strains (permafrost, soil and water) as well as in clinical strains. It was also revealed that the plasmids of the novel group were characterized by a narrow host range; they were distributed among strains belonging to the genus *Pseudomonas* only. 

It should be noted that the genus *Pseudomonas* is very wide and contains phylogenetically distant groups. Our data indicate the possibility of a limited distribution of plasmids related to pPPUT-Tik1-1 among strains belonging to the *P. aeruginosa* group. Indeed, we were unable to transfer pPPUT-Tik1-1 into any of the six used *P. aeruginosa* strains using conjugation. However, in GenBank, there is one complete genome of the plasmid pNY7610-IMP revealed into the strain of this species and four plasmids found among unassembled genomes of *P. aeruginosa*. It is noteworthy that this particular plasmid has a greater similarity of replication genes with a group of short non-conjugative plasmids (Figure 4), which may also explain its ability to replicate in the *P. aeruginosa* strains. Alternatively, some strains of *Pseudomonas* may be misannotated in the database [26,27,33]. Additional research is needed to determine the exact host range of the pPPUT-Tik1-1-related plasmids.

While the molecular structure of the backbone region of pPPUT-Tik1-1-related plasmids is characterized by a high level of homology, the accessory regions of these plasmids are highly diverse. Some contain operons (transposons) of resistance to mercury, and/or modules of resistance to other heavy metals (copper, cobalt/zinc/cadmium, as well as chromium) (Table 1 and Appendix A). Others contain genes of resistance to antibiotics (streptomycin, aminoglycosides and beta-lactams). Some plasmids possess genes encoding resistance to both heavy metals and antibiotics. Finally, there are plasmids with organic solvent degradation genes. Thus, plasmids with different phenotypes belong to the same incompatibility group. Previously, the same pattern was described for *Pseudomonas* plasmids from the IncP-7 group [34] and the IncpRBL16 plasmids [10]. 

Numerous studies have shown a wide distribution of operons and transposons of mercury resistance in bacteria both in natural conditions and in a clinic [35,36,37,38,39]. The cause is believed to be the ubiquitous presence of mercury compounds on earth, as a result of natural phenomena [40,41,42] as well as human activities [41]. According to recently published data, up to 7.75% of bacterial genomes contain the *merA* gene(s), encoding mercury reductase, responsible for resistance to mercury [43]. In plasmids (contigs) of the pPPUT-Tik1-1 group, this number is 60% (27 from 45). The reason for this high frequency of occurrence of mercury resistance transposons among plasmids of the pPPUT-Tik1-1 group remains unclear. Also surprising is the wide distribution of the Tn*5563a* transposon, which contains a defective *mer* operon carrying intracellular mercury transport genes *merP* and *merT* but lacking the mercury reductase gene. It is known that such combination of *mer* genes may cause hypersensitivity to mercury compounds [44,45]. However, in many cases, the genomes of these strains simultaneously contain complete *mer* operons, which could neutralize the negative effect of Tn*5563a* (Table 2 and Appendix A).

The possible role of plasmids of the novel group in the spread of antibiotic resistance in a clinic remains to be studied. Since most *Pseudomonas* strains are rare in a clinic, and do not have high virulence, some researchers suggested that they may play a role in the exchange of antibiotic resistance genes to virulent strains of *P. aeruginosa* [46,47]. Most of the plasmids from the new group do not contain determinants of antibiotic resistance and likely do not play much of a role in the distribution of these genes or their accumulation. However, this year, a complete sequence of a plasmid from a clinical strain of *P. aeruginosa* NY7610 carrying a class 1 integron and six resistance genes to antibiotics of various classes was submitted to GenBank. As in the case of environmental *Acinetobacter* mega-plasmids [48], the active use of antibiotics may result in the recruitment of new environmental plasmids to a clinic. Thus, our data indicate that the problem of combating the global spread of antibiotic resistance necessitates a multidisciplinary, multisector and coordinated approach to address health threats at the human–animal–environment interface, which was formulated in the One Health concept [49]. 

At the same time, the wide presence of pPPUT-Tik1-1-like plasmids in strains inhabiting various ecological niches suggests that they may play a role in adaptability of their host bacteria. This assumption is confirmed by the fact that many pPPUT-Tik1-1-like plasmids encode adaptive genes that are different for different plasmids.

In conclusion, it should be emphasized that the data obtained from the comparative analysis of plasmids of the pPPUT-Tik1-1 group convincingly indicate the need and promise of using complex approaches in the classification of plasmids.

## 4. Materials and Methods

### 4.1. Media and Growth Conditions

Bacteria were grown in a lysogeny broth (LB) medium or solidified agar LB medium (LA) [50] at 30 °C. When required, the LA medium was supplemented with antimicrobial agents at the following final concentrations (µg/mL): HgCl_2_ (Hg), 5–8; streptomycin (Sm), 100–200; and rifampicin (Rif), 25.

### 4.2. Bacterial Strains 

The strain of *P. putida* Tik1 was isolated from a permafrost sample aged 220–390 thousand years collected in the area of the Laptev Sea Coast [51] with all precautions to prevent any contamination of samples and the isolation of exogenous strains as described in [20,52]. A variety of *Pseudomonas* species strains were obtained from the All-Russian National Collection of Industrial Microorganisms (VKPM) and All-Russian Collection of Microorganisms (VKM) and from the collection of the Institute of Molecular Genetics (IMG) (Table 2). All the Pseudomonas strains used as recipients in conjugative crosses were first marked with spontaneous mutations to rifampicin resistance (rifR). To this goal, bacterial cells grown in LB to OD600 = 0.8 were plated onto LA supplemented with rifampicin. The colonies were replated on the same medium. Mutants with the best growth in the presence of rifampicin were selected and used as recipients in mating experiments. 

### 4.3. Mating

Plasmid conjugal transfer experiments were carried out as previously described [48]. Briefly, overnight cultures were diluted 50-fold in fresh broth and were grown for 3–4 h (to the logarithmic growth stage); then, a total of 25 µL of culture of donor strain Tik1 and each of the recipient bacteria (Table 2) were mixed together, plated onto LA and incubated at 30 °C for 16–18 h. The grown cells were scrapped off the plate, resuspended and plated onto LA supplemented with antimicrobial agents after appropriate dilutions. 

### 4.4. Plasmid Stability Assays

Plasmid stability assays were performed essentially as described by Deane and Rawlings [53]. Briefly, transconjugants containing pPPUT-Tik1-1 were grown to the stationary phase in the absence of selection and then diluted 100-fold in fresh broth. This procedure was repeated twice a day for 4 days. The final cultures were plated onto LA after appropriate dilutions, and tested for streptomycin and mercury resistance.

### 4.5. Identification of the Backbone Region of pPPUT-Tik1-1 

The genes involved in conjugation (*mob* genes and mating pair formation (MPF) genes) were identified using amino acid similarity with genes of previously described plasmids. The plasmid R64 (AB027308.1, NC_005014) from *Salmonela enterica* and plasmid pA297-3 (KU744946.1) isolated from *A. baumannii* A297 [54] were used as references for the MPF_I_ group of the T4SS system and CPT4 from the MOB_F_ family. 

### 4.6. Search for Plasmids Related to pPPUT-Tik1-1 in GenBank 

Plasmids related to pPPUT-Tik1-1 from modern *Pseudomonas* strains were identified using the BLASTn program. The sequence of pPPUT-Tik1-1 was used as a query to search for related plasmids in the NCBI database containing complete plasmid genomes on 1 April 2023. All plasmids that had a query cover greater than 50% and an identity of the common region greater than 98.5% were considered as related to pPPUT-Tik1-1. Since all the detected plasmids contained genes encoding the coupling protein TrbC and the putative replication initiator protein Rep, the sequences of these two genes were used as queries to search for related sequences in the NCBI database containing whole-genome shotgun contigs on 31 December 2022. 

### 4.7. Whole-Genome Sequencing and Plasmid Assembly 

The Illumina (Illumina, San Diego, CA, USA) and single molecular nanopore sequencing (Oxford Nanopore, Oxford, UK) technologies were used for the sequencing of the genome of *P. putida* Tik1. As a result of sequencing of the genomic DNA library on Illumina MiSeq, 3.3 million read pairs (2 × 300 bp) were obtained. Additionally, genomic DNA was sequenced using the MinION system (Oxford Nanopore, Oxford, UK), resulting in 65,548 reads with an average length of 19,027 bp (a total of 1.2 Gb). Nanopore reads were assembled with Flye v. 2.7b software. Illumina reads were used to polish the assembly with two iterations using pilon v1.22. As a result, a complete circular sequence of the chromosome with a length of 6,433,697 bp and one circular contig of 153,663 bp, representing the plasmid designated pPPUT-Tik1-1, were obtained. 

### 4.8. Bioinformatic Analysis

The multiple alignment of relaxase genes was constructed with MUSCLE v3.8.31; the alignment was used as an input for the maximum likelihood tree construction in *PhyML* 3.0 with default parameters. All genes encoded by 15 plasmids related to pPPUT-Tik1-1 were clustered using the CD-HIT v4.6 program with a minimum global sequence identity of 50%. Plasmids were clustered according to the presence/absence of genes using the hclust function in the R v4.2.1 programming language; the results of the analysis were visualized with the heatmap.2 function in R. The phylogenetic analysis of RepA1, RepA2 and relaxase MobA was performed using the FastTree v.2.1.11 tool with default parameters; MUSCLE was used to build the input alignments.

## Figures and Tables

**Figure 1 ijms-24-13518-f001:**
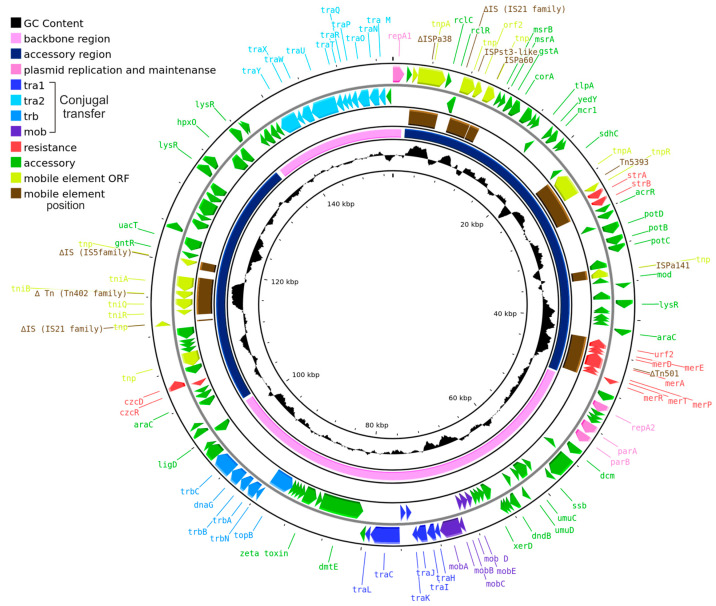
Schematic map of pPPUT-Tik1-1. Genes are denoted by arrows and colored based on the gene function classification shown in the upper left corner. The innermost circle represents GC content. The second circle shows positions of the backbone (blue) and accessory (pink) regions. The third circle shows positions of mobile elements (brown). The fourth and fifth circles show genes encoded in the minus and plus plasmid strands, respectively.

**Figure 2 ijms-24-13518-f002:**
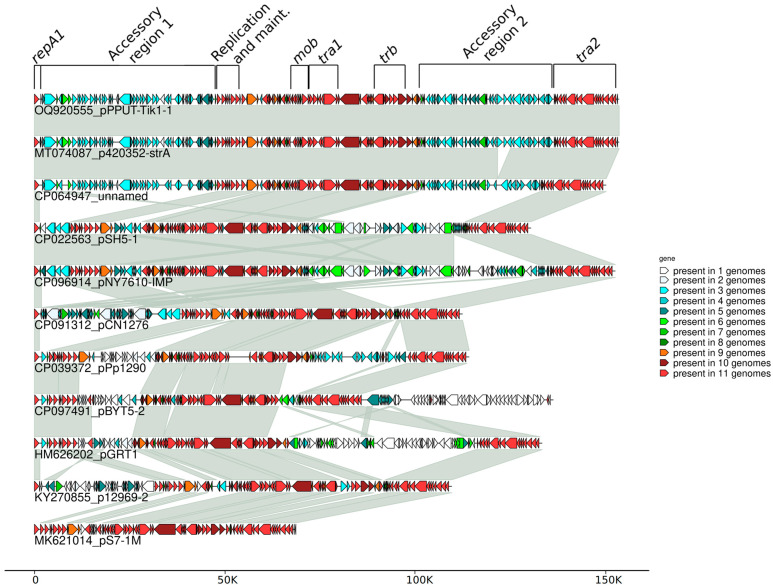
Comparative molecular structure of the plasmids, related to pPPUT-Tik1-1.

**Figure 3 ijms-24-13518-f003:**
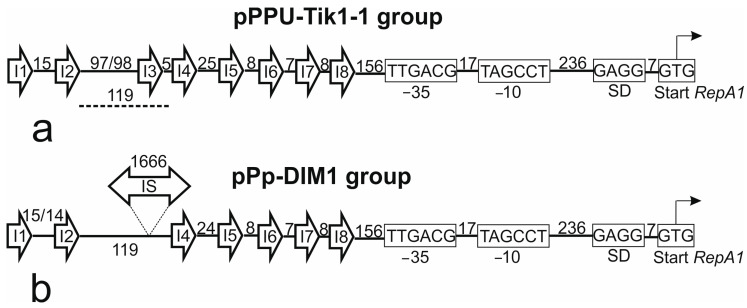
Schematic representation of the iterons and downstream regions of plasmids related to pPPUT-Tik1-1. The number of nucleotides between iterons and conservative elements of the *repA1* gene promoter is indicated. (**a**) Conjugative plasmids. The distance between the second and fourth iterons in plasmid pS7-1M is denoted by dotted line. (**b**) Non-conjugative plasmids.

**Figure 4 ijms-24-13518-f004:**
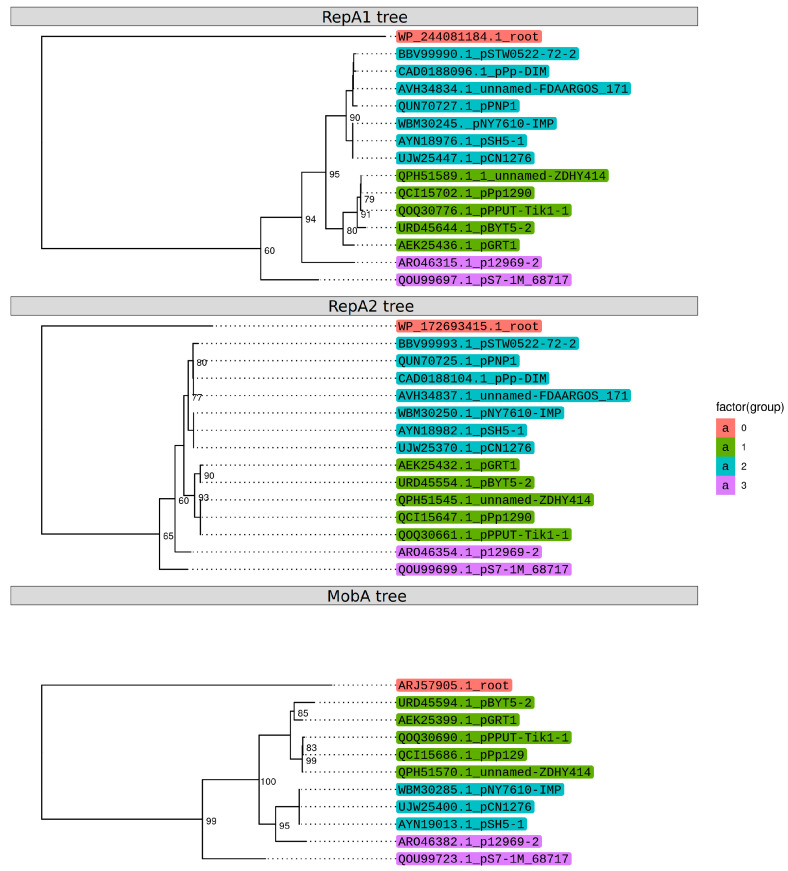
Phylogenetic trees of the RepA1, RepA2 and MobA proteins.

**Table 1 ijms-24-13518-t001:** Characteristics of the *Pseudomonas* plasmids related to pPPUT-Tik1-1.

Strain	Plasmid	Size (bp)	Source	Accessory Region ^1^	Accession Number
Conjugative plasmids (Tra+)					
*P. putida* Tik1	pPPUT-Tik1-1	153,663	permafrost, Russia	Hg, Str	OQ920555
*P. putida* 15420352	p420352-strA	153,678	urine, China	Hg, Str	MT074087
*P. fulva* ZDHY414	Unnamed	150,273	clinic, China	Hg, Str	CP064947
*P. putida* 12969	p12969-2	109,708	clinic, China	Hg, Str, Sul	KY270855
*P. putida* 1290	pPp1290	114,265	pear pyllosphere	Aromatic compounds degradation	CP039372
*P. putida* DOT-T1E	pGRT1	133,451	unknown	Tolerance to toluene, UV-resistance	HM626202
*P. juntendi* 18091276	pCN1276	112,529	urine, China	Hg	CP091312
*P. monteilii* B5	pSH5-1	130,536	soil	Str	CP022562
*Pseudomonas* sp. BYT-5	pBYT5-2	136,400	soil	Molybdate absorption	CP097489
*P. aeruginosa* NY7610	pNY7610-IMP	152,682	clinic, China	Str, Gm, Km, Cb, Cf	CP096914
*P. syringae* pv. actinidiae S7-1M	pS7-1M	68,717	kiwifruit orchard, New Zealand	-	MK621014
Non-conjugative plasmids (Tra−)					
*P. putida*	pPp-DIM	69,823	water, Spain	Sul, Km/Gm, Cb, Tc	LR822050
*P. monteilii*	unnamed	60,588	sputum, USA	-	CP014061
*P. monteilii* STW0522-72	pSTW0522-72-2	59,057	hospital sewage, Japan	-	AP022475
*Pseudomonas* sp.	pPNP1	125,508	activated sludge, USA	Sul, 4-Nitrophenol degradation	CP073662

^1^ Resistance to Hg compounds (Hg), streptomycin (Str), sulphonylamides (Sul), gentamicin (Gm), kanamycin (Km), carbapenems (Cb), cefalosporins (Cf), tetracycline (Tc).

**Table 2 ijms-24-13518-t002:** Frequency of conjugative transfer of pPPUT-Tik1-1 into *Pseudomonas* strains from different phylogenetic groups.

Group ^1^	Strain	Species (Collection ^2^)	Transconjugant Frequency (Per Recipient) ^3^
*P. aeruginosa*	PAO Type	*P. aeruginosa* (IMG)	<10^−9^
	B-6643	*P. aeruginosa* (VKPM)	<10^−9^
	B-5807 Type	*P. aeruginosa* (VKPM)	<10^−9^
	TA37-4	*P. aeruginosa* (IMG)	<10^−9^
	FA8-1	*P. aeruginosa* (IMG)	<10^−9^
	TA40-1	*P. aeruginosa* (IMG)	<10^−9^
*P. resinovorans*	B-14151 Type	*P. resinovorans* (VKPM)	1.9 × 10^−3^
*P. stutzeri*	B-975 Type	*P. stutzeri* (VKM)	2.4 × 10^−6^
	B-903	*P. stutzeri* (VKM)	3.4 × 10^−4^
*P. oleovorans*	B-8623	*P. oleovorans* (VKPM)	9.8 × 10^−5^
*P. straminea*	B-14152 Type	*P. seleniipraecipitans* (VKPM)	<10^−9^
*P. putida*	B-4589 Type	*P. putida* (VKPM)	3.7 × 10^−3^
*P. syringae*	B-1546	*P. savastanoi* (VKM)	1.2 × 10^−3^
*P. fluorescens*	B-6735	*P. fluorescens* (VKPM)	5.7 × 10^−4^
	P22-1-2	*P. fluorescens* (IMG)	2.6 × 10^−4^
	B-14148 Type	*P. fluorescens* (VKPM)	4.7 × 10^−3^

^1^ According to [26,27]. ^2^ All-Russian National Collection of Industrial Microorganisms (VKPM), All-Russian Collection of Microorganisms (VKM), the collection of the Institute of Molecular Genetics (IMG). ^3^ Average of three experiments.

**Table 3 ijms-24-13518-t003:** Mercury transposons found in plasmids from pPPUT-Tik1-1 group.

Strain	Plasmid	*mer* Operon	Transposon
*P. putida* Tik1-1	pPPUT-Tik1-1	*merRTPADE*	ΔTn*501*
*P. putida* 12969	p12969-2	*merRTPABGD*	ΔTn*5041D*
		*merRTPCADE*	Tn*5046*
*P. juntendii* 18091276	pCN1276	*merRTPFADE*	Tn*512*
		*merRTPCADE*	Tn*5046*

## Data Availability

The sequence of pPPUT-Tik1-1 was submitted to GenBank and can be found under the accession number OQ920555.

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
