# Peer review of "Conjugative Plasmid pPPUT-Tik1-1 from a Permafrost Pseudomonas putida Strain and Its Present-Day Counterparts Inhabiting Environments and Clinics"

_ijms, 2023, doi:10.3390/ijms241713518_

Round 1

Reviewer 1 Report

The study by Maslova et al., describes the plasmid biology of pPPUT-Tik1-1 isolated from P. putida with origin in permafrost. The text brings an interesting insight into the biology of this plasmid and compares its structure to recently-isolated plasmids available in GenBank. 

The Results part resembles more discussion since it has numerous references and discusses the results directly. It is not suitable and also appears to be confusing regarding what are the original results obtained within this study and which have been already published.

Major comment:

I was further lacking how was the sampling performed which is connected to my biggest concern; since an almost identical plasmid (p420352-strA) was found in China in a patient, can you fully exclude possible contamination? E.g., by bird faeces or human activity, or as mentioned before - during sampling? I think the "One Health" point of view should be considered too on the top of the statement "The discovery of slightly different variants of the same plasmid in ancient and modern Pseudomonas strains belonging to different species indicates the success of this plasmid and suggests its adaptability to various habitats.

Furthermore, I have minor comments/questions:

Why was conjugation of Pseudomonas plasmids attempted in E. coli and Acinetobacter, is their existence outside of Pseudomonas confirmed?

In chapter "2.4. Determination of the host range of Tik1-1 and its stability in different hosts" 

the title is a bit misleading since you introduced the plasmid via conjugation and hence hosts that did not obtain the plasmid this way could not be examined. Have you tried electroporation too?

Lines: 323-327 have not correctly written the bla gene variants (blaVIM-2 etc..)

Lines 259-262: However, a plasmid pNY7610-IMP, related to pPPUT-Tik1-1, was isolated from P. aeruginosa (CP096914) (see above), and we also found four unassembled P. aeruginosa genomes containing plasmids from this group (Table S3), suggesting that plasmids from this group can also be maintained in other species of Pseudomonas genus.

Could you elaborate in the text on what kind of data is this meant for, where are the assemblies coming from, is pure culture etc.? Because detecting plasmids from unassembled genomes sounds like it can be contamination or metagenomic data etc.

In the abstract, you have mispelled GenBank as "Gene Bank".

Author Response

Comments and Suggestions for Authors

The study by Maslova et al., describes the plasmid biology of pPPUT-Tik1-1 isolated from P. putida with origin in permafrost. The text brings an interesting insight into the biology of this plasmid and compares its structure to recently-isolated plasmids available in GenBank. 

The Results part resembles more discussion since it has numerous references and discusses the results directly. It is not suitable and also appears to be confusing regarding what are the original results obtained within this study and which have been already published.

- We are grateful to the reviewer for his/her interest in our work and useful comments and suggestions. We tried to take all of them into account as much as possible in process of the revision of the manuscript.

We carefully checked the necessity of each reference in the Results part and moved the references 19 and 24 to Discussion (now references # 32, 31). The remaining references correspond to the studies describing the functions of genes found in the plasmid pPPUT-Tik1-1 and related plasmids. We have indicated in each place which data were obtained in this study and what was shown earlier. We believe that these references are necessary for clear presentation of the results and cannot be moved to the Discussion part. We have also preserved a short explanation of the conjugation results (Results, chapter 2.5 :Probably, this may be due to some specific properties of the recipient strain [27]”), since it is not connected to the general Discussion.

Major comment:

I was further lacking how was the sampling performed which is connected to my biggest concern; since an almost identical plasmid (p420352-strA) was found in China in a patient, can you fully exclude possible contamination? E.g., by bird faeces or human activity, or as mentioned before - during sampling? I think the "One Health" point of view should be considered too on the top of the statement "The discovery of slightly different variants of the same plasmid in ancient and modern Pseudomonas strains belonging to different species indicates the success of this plasmid and suggests its adaptability to various habitats.

-  Thank you for your comment. Indeed, it is necessary to have guarantees of the authenticity of the samples and to observe strict measures to exclude the contamination of the samples by modern microorganisms during microbiological studies of permafrost. A group of scientists from various fields led by David Gilichinsky were the first to prove that permafrost is a natural repository of ancient microorganisms. To make this conclusion they developed standard methods of sterile sampling, delivery and storage of the samples, as well as their plating. All these methods were described in the article of Vorobieva et al.(Vorobyova, E, Soina, V, Gorlenko, M, Minkovskaya, N, Zalinova, N, Mamukelashvili, A, Gilichinsky, D, Rivkina, E, Vishnivetskaya, T. The deep cold biosphere: facts and hypothesis. FEMS Microbiol Rev. 1997, 20, 277-90.), and we now have added it to the list of references (52). Two authors of the present work, Drs Mayya Petrova and Sofia Mindlin, have collaborated with Gilichinsky’s research group for many years. Since 1989, in our laboratory we have been conducting comprehensive studies of mobile elements from permafrost involved in the horizontal transfer of genes for resistance to heavy metals and antibiotics. We have found that antibiotic resistance genes, as well as mobile elements involved in their horizontal transfer, were acquired by clinical strains from environmental bacteria. The methods of sampling and bacteria isolation as well as the results of the studies of these strains have been described in detail in more than twenty articles on this topic. Later, a group of Canadian scientists conducted similar studies and confirmed the correctness of our conclusions (D’Costa V.M., King C.E., Kalan L., Morar M., Sung W.W., Schwartz C. et al. Antibiotic resistance is ancient. Nature. 2011; 477(7365): 457—61. doi: 10.1038/nature10388). The strain Tik1 was isolated and described in an earlier study by Mindlin et al., 2009. We have provided this reference in the revised in the relevant place in the Introduction, Methods and Discussion chapters. manuscript and added in the methods. In addition, we have added the information about how the sampling was performed to the Methods chapter and added two more references describing this in detail. (references #51, 52)

We have also mentioned the "One Health" concept in connection with our data in the Discussion.

Furthermore, I have minor comments/questions:

Why was conjugation of Pseudomonas plasmids attempted in E. coli and Acinetobacter, is their existence outside of Pseudomonas confirmed?

- Thank you. Analysis of the host range of the pPPUT-Tik1-1 plasmid was performed earlier, before sequencing of the plasmid itself and of the complete genome of the strain Tik1 (Mindlin et al, 2009). Therefore, in the cited study, the plasmid was briefly examined using classical genetic methods. We have indicated that these data were obtained previously, while in the current study the transfer of pPPUT-Tik1-1 and related plasmids was tested in various Pseudomonas species.

In chapter "2.4. Determination of the host range of Tik1-1 and its stability in different hosts" 

the title is a bit misleading since you introduced the plasmid via conjugation and hence hosts that did not obtain the plasmid this way could not be examined. Have you tried electroporation too?

- Large plasmids are usually broken down during electroporation, so conjugative plasmids are traditionally transferred from cell to cell by conjugation. We have modified the title of the chapter accordingly.

Lines: 323-327 have not correctly written the bla gene variants (blaVIM-2 etc..)

- Thank you. We have corrected these errors.  

Lines 259-262: However, a plasmid pNY7610-IMP, related to pPPUT-Tik1-1, was isolated from P. aeruginosa (CP096914) (see above), and we also found four unassembled P. aeruginosa genomes containing plasmids from this group (Table S3), suggesting that plasmids from this group can also be maintained in other species of Pseudomonas genus.

Could you elaborate in the text on what kind of data is this meant for, where are the assemblies coming from, is pure culture etc.? Because detecting plasmids from unassembled genomes sounds like it can be contamination or metagenomic data etc.

- Thank you. All four plasmids were found in unassembled genomes of four strains of P. aeruginosa , isolated in pure culture. We have indicated this in the revised manuscript: “However, a plasmid pNY7610-IMP, related to pPPUT-Tik1-1, was isolated from P. aeruginosa (CP096914) (see above), and we also found four unassembled genomes of different environmental and clinical P. aeruginosa strains containing related plasmids (Table S3)”. All the information about these strains is also incleded in Table S3.

In the abstract, you have mispelled GenBank as "Gene Bank".

-Thank you. We have corrected this error.

Reviewer 2 Report

Response for the authors

The work is extremely technical, it would fit into a short communication, and thus the amount of information and number of pages is reduced. the information provided is interesting, only that it does not provide precise data regarding the reason why that permafrost was studied, the fact that the strain of Pseudomonas putida was isolated is a proposed fact or simply the permafrost was microbiologically tested, and this was the result. Were other bacteria isolated? Have there been other similar studies, of microbiological analysis of permafrost?

We appreciate the effort of the authors for the submitted activity, but the work requires major revisions to be able to answer some of the questions asked. Some objectives that need to be corrected and described are reproduced below:

....Initially, the classification of plasmids was based on their ability (inability) to coexist in a bacterial cell. Line 39 – requires bibliography

It would be necessary to add more information about Pseudomonas putida, about the known genes, which plasmids it encodes, etc. in the introduction.

Is it a single tested and analyzed strain, Tik 1?

Under what conditions did you end up studying the strain of Pseudomonas putida in permafrost? Did you look for this bacterial species on purpose or was it simply an accident that you isolated this bacterial species?

What is the importance of the obtained results? Does it have any significance in relation to Pseudomonas species today or is it just a discovery of the presence of a strain of Pseudomonas putida in the permafrost, which has been preserved over time. Can these plasmids be responsible for antibiotic resistance?

Line 427: correct to degrees Celsius.

The conclusion chapter is missing from the manuscript.

Author Response

The work is extremely technical, it would fit into a short communication, and thus the amount of information and number of pages is reduced. the information provided is interesting, only that it does not provide precise data regarding the reason why that permafrost was studied, the fact that the strain of Pseudomonas putida was isolated is a proposed fact or simply the permafrost was microbiologically tested, and this was the result. Were other bacteria isolated? Have there been other similar studies, of microbiological analysis of permafrost?

- Thank you for the thorough reading of our manuscript and for your valuable comments and questions. We started to study resistance genes and mobile genetic elements of permafrost bacteria many years ago, when it was proved that permafrost is a natural repository of bacterial communities that existed long before the beginning of active anthropogenic impact on the biosphere and, in particular, before the beginning of the "antibiotic era" (Vorobyova, E, Soina, V, Gorlenko, M, Minkovskaya, N, Zalinova, N, Mamukelashvili, A, Gilichinsky, D, Rivkina, E, Vishnivetskaya, T. The deep cold biosphere: facts and hypothesis. FEMS Microbiol Rev. 1997, 20, 277-90.). During following years, comparative studies of the resistance determinants of permafrost and modern bacteria has given us the opportunity to answer many questions concerning the origin, distribution and evolution of these determinants. In particular, we have found that antibiotic resistance genes, as well as mobile elements involved in their horizontal transfer, were obtained by clinical strains from environmental bacteria. These results have demonstrated that permafrost is an invaluable source of ancient environmental and have been published in two dozen articles on this topic, which are not cited here. We have explained the reasons for studying the permafrost strain in the last paragraph of Introduction. We believe that the results obtained in this work and the range of problems that we are discussing require a full-length article format and cannot be presented in the form of a short communication.

 We appreciate the effort of the authors for the submitted activity, but the work requires major revisions to be able to answer some of the questions asked. Some objectives that need to be corrected and described are reproduced below:

 ....Initially, the classification of plasmids was based on their ability (inability) to coexist in a bacterial cell. Line 39 – requires bibliography

- Thank you. We have added references to the relevant stidies (references ## 5, 6, 7).

It would be necessary to add more information about Pseudomonas putida, about the known genes, which plasmids it encodes, etc. in the introduction.

- Thank you for this suggestion. In this study, we describe a new group of plasmids that are found in strains belonging to different Pseudomonas species, not only to Pseudomonas putida. In addition, we discuss possible new approaches to classification of plasmids. Accordingly, the Introduction is devoted to describing the state of the problem on these topics and the goals of this study. In response to your comment, we have added a short characteristic of Pseudomonas putida into the Introduction (last paragraph): “P. putida is a ubiquitous rhizosphere saprophytic bacterium and soil colonizer that belongs to the wide group of fluorescent Pseudomonas species [21]”. Since comparative analysis of all Pseudomonas plasmids was not a goal of our study, we have only briefly mentioned the diversity of Pseudomonas plasmids in the Introduction: “Many Pseudomonas isolates harbour plasmids, which contribute to the adaptability of Pseudomonas species in a variety of natural habitats”.

Is it a single tested and analyzed strain, Tik 1?

- Thank you for your question. We have indicated that the strain Tik1 was isolated previously, and provided a reference to the previous publication, describing all the details about this sample (last paragraph of Introduction). This work is part of many years of research on resistance genes and mobile genetic elements found in permafrost bacteria (see our response to the first comment above). Another strain of Pseudomonas containing the transposon Tn5045, was also isolated from the same permafrost sample, as was described by us earlier (Petrova M., Gorlenko Zh., Mindlin S. 2011. Tn5045, a novel integron-containing antibiotic and chromate resistance transposon isolated from a permafrost bacterium. Res Microbiol, 162, 337-345). In total, we analyzed more than two hundred samples, isolated in pure cultures about one hundred strains and about half of them were described in our earlier publications.

Under what conditions did you end up studying the strain of Pseudomonas putida in permafrost? Did you look for this bacterial species on purpose or was it simply an accident that you isolated this bacterial species?

 - Thank you this question. We have provided the reference to the previous study describing isolation of this strain, in the last paragraph of Introduction. The media we use, LA, R2A and TSA, allow us to isolate most common soil strains, mainly belonging to the genera Pseudomonas, Acinetobacter, Stenotrophomonas, Xanthomonas, Psychrobacter, Brevundimonas, Bacillus, Arthrobacter, Micrococcus. We deliberately do not use special reactivation or cumulative methods, since such methods can cause contamination. As a rule, gram-positive bacteria mainly regrow from permafrost samples. The strain Tik1 of Pseudomonas putida analyzed in this study was initially selected for further analysis based on its resistance to Hg and Str, suggesting that it may contain specific resistance determinants.

What is the importance of the obtained results? Does it have any significance in relation to Pseudomonas species today or is it just a discovery of the presence of a strain of Pseudomonas putida in the permafrost, which has been preserved over time. Can these plasmids be responsible for antibiotic resistance?

- Thank you for these comments. The main novelty of our results is in the discovery of a novel group of large conjugative plasmids, which contain various accessory genes with adaptive functions and are spread in environmental and clinical strains of Pseudomonas. We have characterized the structure of these plasmids in detail and analyzed the conservation and diversity of their backbone and accessory regions. We have indicated this in the revised manuscript (Abstract, Discussion). The results further suggest that environmental strains are the source of clinical resistance genes and mobile elements involved in their distribution, and that these genes and mobile elements were distributed long before the beginning of active anthropogenic activity. In particular, the plasmid pPPUT-Tik1-1 contains streptomycin resistance genes strA-strB, and two modern plasmids, almost identical to pPPUT-Tik1-1, also contain these genes. In addition, we found other related plasmids with other antibiotic resistance genes in the database (Tables 1 and S3). We have discussed the possible role of this group of plasmids in the spread of resistance genes among clinical strains in Discussion. These data are also very important in connection with of One Health concept, which we have discussed at the request of another reviewer.

Line 427: correct to degrees Celsius.

 - Thank you. This has been done.

The conclusion chapter is missing from the manuscript.

- The rules of the journal allow the authors to submit  manuscripts lacking the Conclusions chapter. We have outlined the main conclusions of the study at the end of Discussion.

Reviewer 3 Report

Maslova et al. describe the structure of a plasmid (here named Tik1) found in a Pseudomonas putida strain isolated from a permafrost sample and compared it with plasmids of Pseudomonas clinical strains A characteristic feature of these plasmids are two unique gene sequences that encode  initiators of replication. Comparison with a wider range of Tik1 plasmids revealed a large variety of different accessory regions. The sequence of the backbone region of the Tik1 plasmid s shows a high degree of homology while the accessory regions are very diverse. The data support the allocation of the Tik1 type plasmids to a new group.

Line 21. Replace functional burden by functions encoded by the accessory regions.

Line 263 Delete in

satisfactory

Author Response

Comments and Suggestions for Authors

Maslova et al. describe the structure of a plasmid (here named Tik1) found in a Pseudomonas putida strain isolated from a permafrost sample and compared it with plasmids of Pseudomonas clinical strains A characteristic feature of these plasmids are two unique gene sequences that encode  initiators of replication. Comparison with a wider range of Tik1 plasmids revealed a large variety of different accessory regions. The sequence of the backbone region of the Tik1 plasmid s shows a high degree of homology while the accessory regions are very diverse. The data support the allocation of the Tik1 type plasmids to a new group.

- Thank you for your appreciation of our manuscript and for your corrections

.Line 21. Replace functional burden by functions encoded by the accessory regions.

 - Thank you. It has been done.

Line 263 Delete in  

- Thank you. Thank you. It has been done.

Round 2

Reviewer 2 Report

I appreciate the effort of the collective of authors in order to improve the quality of the manuscript. They tried and managed to answer all the questions. I recommend publishing the manuscript in revised form.
